# Towards Demystifying the Generalization Behaviors When Neural Collapse Emerges

## Abstract

Neural Collapse (NC) is a well-known phenomenon of deep neural networks in the terminal phase of training (TPT). It is characterized by the collapse of features and classifier into a symmetrical structure, known as simplex equiangular tight frame (ETF). While there have been extensive studies on optimization characteristics showing the global optimality of neural collapse, little research has been done on the generalization behaviors during the occurrence of NC. Particularly, the important phenomenon of generalization improvement during TPT has been remaining in an empirical observation and lacking rigorous theoretical explanation. In this paper, we establish the connection between the minimization of CE and a multi-class SVM during TPT, and then derive a multi-class margin generalization bound, which provides a theoretical explanation for why continuing training can still lead to accuracy improvement on test set, even after the train accuracy has reached 100%. Additionally, our further theoretical results indicate that different alignment between labels and features in a simplex ETF can result in varying degrees of generalization improvement, despite all models reaching NC and demonstrating similar optimization performance on train set. We refer to this newly discovered property as *"non-conservative generalization"*. In experiments, we also provide empirical observations to verify the indications suggested by our theoretical results.

## 1 Introduction

Deep learning models have achieved tremendous success across a wide range of applications. In the context of classification problems, a typical deep model comprises a deep feature extractor along with a linear classifier appended at the end of the extractor. Recently, Papyan et al. (2020) discovered the phenomenon of *neural collapse* (NC) with respect to the last-layer extracted feature and the linear classifier. Some appealing properties with the collapsed within-class feature representation and the maximized equiangular separation among feature centers of different classes emerge after a model has achieved perfect classification on train set while the train loss is still above 0, known as the *terminal phase of training* (TPT). During TPT, it is also observed that the model robustness and generalization on test set are also getting improved (Papyan et al., 2020).

The empirical observation of neural collapse has attracted a lot of following studies to unveil the physics behind such a phenomenon. Under a simplified model, it is shown that the optimization with the widely used loss functions, such as the cross-entropy (CE) loss and the mean squared error (MSE) loss, converge to neural collapse with weight decay regularizer (Zhu et al., 2021; Han et al., 2022; Tirer & Bruna, 2022) and feature constraints (Fang et al., 2021; Graf et al., 2021; Lu & Steinerberger, 2022; Yaras et al., 2022b). Other than investigating the global optimizer, some studies also reveal the benign optimization landscape for the CE (Ji et al., 2022; Zhu et al., 2021; Yaras et al., 2022b) and MSE (Zhou et al., 2022a) loss functions. All these studies try to theoretically explain the occurrence of neural collapse by characterizing the optimization on train set, however, leave the underlying generalization behaviors not yet understood. Whereas an improved test accuracy can be observed in TPT, there is very limited research conducted on the generalization analysis of a model exhibiting NC. Galanti et al. (2022) demonstrate how a classifier generalizes to unseen samples through the lens of NC. However, a theoretical support for the improved generalization during TPT in which NC emerges has not been rigorously established (Hui et al., 2022; Kothapalli et al., 2022). To this end, in this paper, we study the following questions.

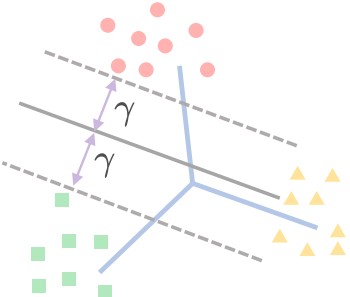 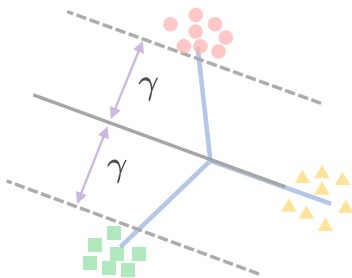

Left: at the beginning of TPT, every sample in the train set starts to be classified correctly, and the margin $\gamma$ between two classes (green and blue) still has potential to be maximized.

Right: during TPT, as the minimization of CE, the margin $\gamma$ is gradually increasing until all features converge to their class centers.

Figure 1: These figures show the distribution of feature in the last layer during NC. The classification problem involves 3 classes, and the feature space is 2-dimensional. We use different color to represent the feature of different classes. The black solid line indicates the decision boundary of binary classification while the dashed lines indicate the supporting hyperplane of features.

*Question 1: Why does the model still exhibits performance improvements on the test set if we continue to train it, even after achieving perfect classification accuracy on all the samples in the training set?*

In Section 4.1, we explore this question to look for the underlying mechanism that can theoretically elucidates the phenomenon. Our results indicate that the NC phenomenon comes from maximizing the minimal margin between any two classes, and we derive a generalization bound that becomes tighter as this margin increases. Concretely, we establish a limit equivalence between the minimization of CE and the multi-class SVM. We prove that in the stage of TPT when CE loss tends to zero, which also means that the feature in the last layer starts to be linearly separable, the linear classifier behaves as a hard-margin multi-class SVM. We then propose a multi-class margin generalization bound based on the original binary classification margin bound, which can help to shed the light on the generalization behaviors during the occurence of NC. When accuracy starts to become 100% on the train set, *i.e.*, the beginning of TPT, the different class centers have approximately formed a simplex equiangular tight frame (ETF). While selecting the class with the maximal logit can predict the correct category for any samples in train set at this moment, the feature learnt by model has not achieved a complete within-class variability collapse, as shown in Figure 1-(left), which means there is still potential such that the margins can be maximized if we continue to train the model. As the CE loss keeps being minimized, the support vectors (the closest features to the prediction boundary) are pushed away from the decision plane to be closer to its class center. Finally when neural collapse emerges, the last-layer feature is collapsed and the minimal pair-wise margin is maximized as shown in Figure 1-(right). Based on our generalization bound, we know that the model with a larger margin enjoys a better generalization performance. This theoretically explains the generalization improvement during TPT, which has only been an important empirical observation in neural collapse (Papyan et al., 2020).

Although training properties are invariant for a model exhibiting NC, many random factors in the modern training paradigm, such as weight initialization and data augmentation, can lead to different optimization path (gradient flow), as a result, the feature and classifier solution structure may converge to simplex ETFs with different directions and permutations. Clearly, the models converging to different simplex ETFs still enjoy the minimal CE and the same NC properties, however, there is no reason to believe that these models will also have the same generalization performance. Thus we are further curious about whether the generalization behaviors will be influenced by the variability of the converged solution structure.

*Question 2: Would a model converging to different simplex ETFs still leads to the same generalization performance? And if not, then why?*

In Section 4.2, we first propose a series of generalization analysis offering the theoretical insight about how generalization could possibly be affected by representation learning with different equiv-

alent simplex ETF (permutation and rotation equivalence, see Definition 4.7). Our proof mainly draws inspiration from an analytical study on the supervised manifold learning for classification (Vural & Guillemot, 2017), which uses the out-of-sample interpolation technique. However, due to the curse of dimensionality, interpolation in high-dimensional data space demands a large sample size, nearly unachievable in real-life situations. Thus we further develop an improved proof scheme, which requires less and weaker assumptions and leads to an error bound with faster convergence of sample size, to analyze the generalization in neural collapse. Our results indicate that different permutations for the solution can cause changed margin and thus affect the generalization bound. We conduct extensive experiments in Section 5 to provide empirical support for our theoretical findings considering both permutation and rotation. In experiments, we find that the models converging to simplex ETFs with different permutations and rotations, which would lead to different alignments between labels and features and different feature directions, have very different test accuracy, even if they have an accuracy of 100% and an almost minimized CE loss on train set. All these theoretical and empirical results indicate that the generalization performance is sensitive to the solution variability. We refer to this intriguing generalization phenomenon as *"non-conservative generalization"*, as different optimization paths, despite showing the same neural collapse properties at convergence, leads to different behaviors with respect to generalization.

In summary, the contributions of this study can be listed as follows:

- We show that the minimization of CE loss can be seen as a multi-class SVM in the last layer during TPT, and derive a multi-class margin generalization bound, which theoretically explains the generalization improvement during the occurrence of neural collapse.
- Inspired by out-of-sample interpolation technique, we conduct a generalization analysis, and develop an improved proof scheme, which requires less assumptions and leads to an error bound with faster convergence. These analysis indicate that the solution variability of a model converging to neural collapse can cause different generalization performance.
- We conduct a series of experiments to provide solid empirical support to our theoretical results. Particularly, we verify the existence of non-conservative generalization in experiments that is suggested by our theoretical analysis.

## 2 RELATED WORK

**Neural Collapse.** Neural Collapse (NC) first observed by Papyan et al. (2020) has inspired numerous studies to theoretically investigate such an appealing phenomenon. Many studies (Ji et al., 2022; Fang et al., 2021; Zhu et al., 2021; Zhou et al., 2022b; Yaras et al., 2022a) have proposed various optimization models and proved the global optimality satisfying neural collapse. For example, Zhu et al. (2021) proposed the unconstrained feature model and provided optimization analysis for NC, and Fang et al. (2021) proposed the layer-peeled model, a nonconvex yet analytically tractable optimization model, to prove the NC optimality and predict the changed phenomenon in imbalanced training case. Other studies have explored NC under the mean squared error (MSE) loss (Han et al., 2022; Tirer & Bruna, 2022; Zhou et al., 2022a; Mixon et al., 2020; Poggio & Liao, 2020). In addition to MSE loss, Zhou et al. (2022b) extended such results and analysis to a broad family of loss functions including the commonly used label smoothing and focal loss. However, these studies can only explain why NC appears on train set, leaving the generalization performance during the occurrence of NC unconsidered. The NC phenomenon has also inspired methods for application problems, such as imbalanced classification (Xie et al., 2022; Yang et al., 2022; Peifeng et al., 2023; Liu et al., 2023) and incremental learning (Yang et al., 2023). Nonetheless, the generalization behaviors of a model converging to NC have not been rigorously understood.

**Generalization Analysis.** Generalization has been an important factor for deep neural networks (Belkin et al., 2019; Soudry et al., 2018; Zhang et al., 2021), and has been studied through the lens of VC dimension (Vapnik & Chervonenkis, 2015; Neyshabur et al., 2017) and the connection with algorithm stability or robustness (Bousquet & Elisseeff, 2002; Xu & Mannor, 2012). There are some studies that focus on the transfer learning ability brought by NC (Hui et al., 2022; Kothapalli et al., 2022; Li et al., 2022) via empirical investigations. Galanti et al. (2022) provide a theoretical framework that shows that neural collapse helps to generalize to new samples and even unseen classes. However, a rigorous explanation for the generalization behaviors during the occurrence of neural collapse still remains to be explored.

## 3 PRELIMINARY

Papyan et al. (2020) conducted extensive experiments to reveal the NC phenomenon on class balanced datasets. This phenomenon occurs during the terminal phase of training (TPT), which starts from the epoch that the training accuracy has reached $100\%$. During TPT, training error rate is zero, but CE loss still keeps decreasing. To describe this phenomenon more clearly, we introduce several necessary notations first. We denote the class number as $C$ and feature dimension as $d$. Here, we consider classifiers with the form $logit = \boldsymbol{M}\boldsymbol{z} = [\langle M_1, \boldsymbol{z}\rangle, \ldots, \langle M_C, \boldsymbol{z}\rangle]^T$, where $\boldsymbol{M} \in \mathbb{R}^{d \times C}$ is the linear classifier, and $\boldsymbol{z}$ is the feature of a sample obtained from a deep feature extractor. The classification result is given by selecting the maximum score of $logit$. Given a balanced dataset, we denote the feature of $i$-th sample in $y$-th category as $\boldsymbol{z}_{y,i}$. Specifically, when the model is trained on a balanced dataset, its last layer would converge to the following manifestations:

**NC1** **Variability Collapse.** All samples belonging to the same class converge to the class mean: $\|\boldsymbol{z}_{y,i} - \bar{\boldsymbol{z}}_y\| \to 0, \forall y, \forall i$ where $\bar{\boldsymbol{z}}_y = \text{Ave}_i\left(\boldsymbol{z}_{y,i}\right)$ denote the class-center of the $y$-th class;

**NC2** **Convergence to Self Duality.** The samples and classifier belonging to the same class converge to duality: $\|\boldsymbol{z}_{y,i} - M_y\| \to 0, \forall y, \forall i$;

**NC3** **Convergence to Simplex ETF.** The classifier weight converges to the vertices of a simplex ETF;

**NC4** **Nearest Class-Center Classification.** The class prediction can be performed by the distance to the nearest class center, *i.e.* $\arg\max_y \langle M_y, z\rangle \to \arg\min_y \|z - \bar{\boldsymbol{z}}_y\|$.

In **NC3**, Simplex ETF is an interesting structure. Note that there exist two different objects with this notion: simplex ETF and ETF. ETF is rooted from frame theory, while Papyan et al. (2020) introduced the simplex ETF in the context of the NC phenomenon with the following definition.

**Definition 3.1** (**Simplex Equiangular Tight Frame** Papyan et al. (2020)). A simplex ETF is a collection of vertices in $\mathbb{R}^C$ specified by the columns of

$$\boldsymbol{M}^{\star} = \alpha R \sqrt{\frac{C}{C-1}} \left(I - \frac{1}{C}\mathbb{I}\mathbb{I}^T\right),$$

where $I \in \mathbb{R}^{C \times C}$ is the identity matrix, $\mathbb{I} \in \mathbb{R}^C$ is the all-one vector, $R \in \mathbb{R}^{d \times C}(d \geq C)$ is an orthogonal projection matrix, $\alpha \in \mathbb{R}$ is a scale factor.

The simplex ETF has a symmetric structure. One can find that the angles between any two vertices in a simplex ETF are equal (the *equiangular* property).

## 4 THEORETICAL RESULTS

**Notations.** Denote feature dimension of the last layer as $d$ and class number as $C$. Suppose sample space is $\mathcal{X} \times \mathcal{Y}$, where $\mathcal{X}$ is data space and $\mathcal{Y} = \{1, \ldots, C\}$ is label space. We assume class distribution is $\mathcal{P}_{\mathcal{Y}} = [p(1), \ldots, p(C)]$, where $p(c)$ denotes the proportion of class $c$. Let the training set $S = \{(\boldsymbol{x}_i, y_i)\}_{i=1}^N$ be drawn i.i.d from probability $\mathcal{P}_{\mathcal{X} \times \mathcal{Y}}$. For $y$-th class samples in $S$, we denote $S_y = \{\boldsymbol{x}|(\boldsymbol{x}, y) \in S\}$ and $|S_y| = N_y$. The form of classifiers is $logit = \boldsymbol{M}^T f(\boldsymbol{x}; \boldsymbol{w}) = [\langle M_1, f(\boldsymbol{x}; \boldsymbol{w})\rangle, \ldots, \langle M_C, f(\boldsymbol{x}; \boldsymbol{w})\rangle]$, where $\boldsymbol{M} \in \mathbb{R}^{d \times C}$ is the last-layer linear classifier, and $f(\cdot; \boldsymbol{w}) \in \mathbb{R}^d$ is the feature extractor parameterized by $\boldsymbol{w}$.

### 4.1 GENERALIZATION WHEN NEURAL COLLAPSE EMERGES

This section mainly answers the ***Question 1***. As we discussed in Section 1, we argue that the NC phenomenon is closely linked with the margin of classification. Therefore, our analysis focuses on margin and is two-fold. First, we find that the minimization of CE during TPT can actually be seen as a multi-class SVM. Then, we focus on the margin-based generalization analysis, which could explain the intriguing generalization behavior during NC.

First, we illustrate the landscape in the last layer during TPT by analyzing the unconstrained feature model (UFM).

**Unconstrained Feature Model.** Following Zhu et al. (2021), we directly optimize sample features to simplify the analysis. It is reasonable because modern deep models are highly over-parameterized to be able to align a feature with any direction. We use $\boldsymbol{z}_{y,i}$ to represent the feature of the $i$-th sample in the $y$-th class.

$$\min_{\boldsymbol{Z},\boldsymbol{M}} \text{CELoss}(\boldsymbol{M},\boldsymbol{Z}) := -\sum_{y=1}^{C}\sum_{i=1}^{N_y} \log \frac{\exp\left(\langle M_y, \boldsymbol{z}_{y,i}\rangle\right)}{\sum_{y'}\exp\left(\langle M_{y'}, \boldsymbol{z}_{y,i}\rangle\right)}. \tag{1}$$

**Theorem 4.1** (**Multiclass SVM**). *For the CE loss function (1), consider a path of gradient flow* $\{(\boldsymbol{M}^{(t)}, \boldsymbol{Z}^{(t)})\}_t$. *If* $\text{CELoss}(\boldsymbol{M}^{(t)}, \boldsymbol{Z}^{(t)}) \to 0$ $(t \to \infty)$, *then* $p_{min} \to \infty$ $(t \to \infty)$, *where* $p_{min}$ *is the margin of the train set*

$$p_{min} := \min_{y \neq y'} \min_{i \in [N/C]} \langle M_y - M_{y'}, z_{y,i}\rangle.$$

**Remark 4.2.** In fact, consider the problem $\max_{\boldsymbol{M},\boldsymbol{Z}} p_{min}$. If we transform the inner $\min$ as the constraint, we can obtain the following multiclass SVM

$$\max_{\boldsymbol{M},\boldsymbol{Z}} \gamma$$
$$s.t. \langle M_y - M_{y'}, z_{y,i}\rangle \geq \gamma, \forall y \neq y', \forall i.$$

This shows that during TPT, the linear classifier works as a potential multi-class SVM under the effect of CE. And the margin of the entire train set is still improving as the CE loss keeps decreasing, which we also demonstrate empirically in Figure 3.

Then, we provide the definition of margin between two classes when features in the last layer are linearly separable:

**Definition 4.3** (**Linear Separability**). Given the dataset $S$ and a classifier $(\boldsymbol{M}, f(\cdot; \boldsymbol{w}))$, if the classifier can achieve $100\%$ accuracy on train set, $\boldsymbol{M}$ has to be able to linearly separate for the feature of $S$, *i.e.*, for any two classes $y, y'(y \neq y')$, there exists a $\gamma_{y,y'} > 0$ such that:

$$(M_y - M_{y'})^T f(\boldsymbol{x}; \boldsymbol{w}) \geq \gamma_{y,y'}, \qquad \forall(\boldsymbol{x}, y) \in S,$$
$$(M_y - M_{y'})^T f(\boldsymbol{x}; \boldsymbol{w}) \leq -\gamma_{y,y'}, \quad \forall(\boldsymbol{x}, y') \in S.$$

In this case, we say the classifier $\boldsymbol{M}$ can linearly separate the features $\{(f(\boldsymbol{x}; \boldsymbol{w}), y)|(\boldsymbol{x}, y) \in S\}$ by margin $\{\gamma_{y,y'}\}_{y \neq y'}$.

It is reasonable to make an assumption that features in the last layer are linearly separable, since this exactly is the scene where NC begins to appear. Then we propose the Multiclass Margin Bound.

**Theorem 4.4** (**Multiclass Margin Bound**). *Consider a dataset $S$ with $C$ classes. For any classifier* $(\boldsymbol{M}, f(\cdot; \boldsymbol{w}))$, *we denote its margin between $y$ and $y'$ classes as* $(M_y - M_{y'})^T f(\cdot; \boldsymbol{w})$. *And suppose the function space of the margin is* $\mathcal{F} = \{(M_y - M_{y'})^T f(\cdot; \boldsymbol{w})|\forall y \neq y', \forall \boldsymbol{M}, \boldsymbol{w}\}$, *whose uppder bound is*

$$\sup_{y \neq y'} \sup_{\boldsymbol{M}, \boldsymbol{w}} \sup_{\boldsymbol{x} \in \mathcal{X}} \left|(M_y - M_{y'})^T f(\boldsymbol{x}; \boldsymbol{w})\right| \leq K.$$

*Then, for any classifier $(\boldsymbol{M}, f(\cdot; \boldsymbol{w}))$ with margins $\{\gamma_{y,y'}\}_{y \neq y'}(\gamma_{y,y'} > 0)$ on the given dataset, the following inequality holds with probability at least $1 - \delta$,*

$$\mathbb{P}_{x,y}\left(\max_c [Mf(\boldsymbol{x}; \boldsymbol{w})]_c \neq y\right) \lesssim \underbrace{\sum_{y=1}^{C} p(y) \sum_{y' \neq y} \frac{\mathfrak{R}_{N_y}(\mathcal{F})}{\gamma_{y,y'}}}_{(\boldsymbol{A})} + \underbrace{\sum_{y=1}^{C} p(y) \sum_{y' \neq y} \sqrt{\frac{\log(\log_2 \frac{4K}{\gamma_{y,y'}})}{N_y}}}_{(\boldsymbol{B})} + L_{0,1}$$

*where $\lesssim$ means we omit probability related terms, and $L_{0,1}$ denotes the empirical risk term:*

$$L_{0,1} = \sum_{y=1}^{C} p(y) \sum_{y' \neq y} \sum_{x \in S_y} \frac{\mathbb{I}((M_y - M_{y'})^T f(x) \leq \gamma_{y,y'})}{N_y}.$$

$\mathfrak{R}_{N_i}(\mathcal{F})$ *is the Rademacher complexity (Kakade et al., 2008; Bartlett & Mendelson, 2002) of function space $\mathcal{F}$. Please refer to Appendix C for full version of this theorem.*

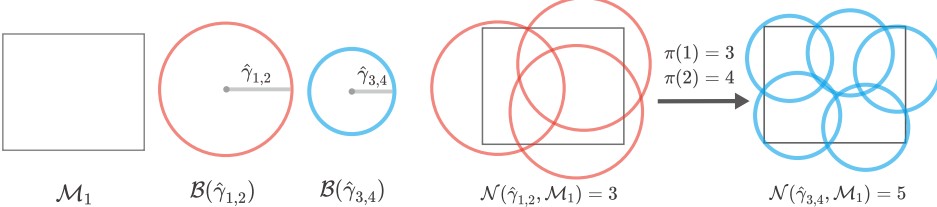

Figure 2: An illustrative example for Remark 4.9, where we denote $\hat{\gamma}_{y,y'} = \frac{\gamma_{y,y'}}{L\|M_y - M_{y'}\|}$ and $\mathcal{B}(\epsilon)$ as a ball of radius $\epsilon$. As shown in the figure, $\hat{\gamma}_{\pi(1),\pi(2)}$, *i.e.*, $\hat{\gamma}_{3,4}$, is smaller than $\hat{\gamma}_{1,2}$, then $\mathcal{M}_1$ requires a larger number of balls to cover it than the one before permutation.

**Corollary 4.5.** *Recall NC occurs when class distribution is uniform, so we let $p(y) = \frac{1}{C}$ and $N_y = \frac{N}{C}$, $\forall y \in [C]$. The generalization bound in Theorem 4.4 then becomes*

$$(\boldsymbol{A}) + (\boldsymbol{B}) = \frac{\mathfrak{R}_{N/C}(\mathcal{F})}{C} \sum_{y=1}^{C} \sum_{y' \neq y} \frac{1}{\gamma_{y,y'}} + \frac{1}{\sqrt{NC}} \sum_{y=1}^{C} \sum_{y' \neq y} \sqrt{\log(\log_2 \frac{4K}{\gamma_{y,y'}})} + L_{0,1}$$

**Remark 4.6** (**The reason for the generalization improvement when NC emerges**). The Multi-class Margin Bound can provide an explanation for the steady improvement in test accuracy during TPT (as shown in Figure.8 and Table.1 of Papyan et al. (2020)). At the beginning of TPT, the accuracy over the training set reaches 100% and $L_{0,1} = 0$, indicating that generalization performance can no longer improve by reducing $L_{0,1}$. However, as we note in Theorem 4.1, $p_{min}$ is still raising if we continue training at this point. And then the margin $\gamma_{i,j}$ also keeps increasing, since $\gamma_{i,j} \geq p_{min}(\forall i \neq j)$. Furthermore, the above two terms related to margin continue to decrease, leading to a better generalization performance.

## 4.2 NON-CONSERVATIVE GENERALIZATION

Then we deal with the ***Question 2***. Our theoretical results suggest that the models that converge to the structure of simplex ETFs with different permutations may exhibit different generalization performances, even if they all have achieved the best performance on the train set and reached neural collapse. First, we introduce the equivalences of simplex ETFs (Holmes & Paulsen, 2004; Bodmann & Paulsen, 2005):

**Definition 4.7** (**Equivalent ETF**). Given two Simplex ETFs $\{\zeta_i\}_{i=1}^{C}, \{\chi_i\}_{i=1}^{C}$ in $\mathbb{R}^d$, they are *Permutation Equivalent* if there exists a permutation matrix $P \in \mathbb{R}^C$ such that $[\zeta_i]_{i=1}^{C} = [\chi_i]_{i=1}^{C} P$; and *Rotation Equivalent* if there exists a orthogonal matrix $R \in \mathbb{R}^d$ such that $[\zeta_i]_{i=1}^{C} = R[\chi_i]_{i=1}^{C}$.

Inspired by the out-of-sample generalization problem (Vural & Guillemot, 2017), we derive the following results.

**Theorem 4.8.** *Given a balanced dataset $S$ and a classifier $(\boldsymbol{M}, f(\cdot; \boldsymbol{w}))$, suppose $(\boldsymbol{M}, f(\cdot; \boldsymbol{w}))$ can linearly separate $S$ by margin $\{\gamma_{y,y'}\}_{y \neq y'}$. Besides, we make the following assumptions:*

- *$f(\cdot, \boldsymbol{w})$ is L-Lipschitz for any $\boldsymbol{w}$, i.e. $\forall \boldsymbol{x}_1, \boldsymbol{x}_2, \|f(\boldsymbol{x}_1, \boldsymbol{w}) - f(\boldsymbol{x}_2, \boldsymbol{w})\| \leq L\|\boldsymbol{x}_1 - \boldsymbol{x}_2\|$*

- *$S$ is large enough such that $N_y \geq \max_{y' \neq y} \mathcal{N}(\frac{\gamma_{y,y'}}{L\|M_y - M_{y'}\|}, \mathcal{M}_y)$ for every class $y$*

- *The tight support of probability $\mathcal{P}_{\boldsymbol{x}|y}$ is denoted as $\mathcal{M}_y$*

*where $\mathcal{N}(\cdot, \mathcal{M}_y)$ is the covering number of $\mathcal{M}_y$. Please refer to Appendix D for its definition. Then the expected accuracy of $(\boldsymbol{M}, f(\cdot; \boldsymbol{w}))$ over the entire distribution is given by*

$$\mathbb{P}_{\boldsymbol{x},y}\left(\max_{y'}[Mf(\boldsymbol{x}; \boldsymbol{w})]_{y'} = y\right) > 1 - \frac{1}{2N} \sum_{y=1}^{C} \max_{y' \neq y} \mathcal{N}(\frac{\gamma_{y,y'}}{L\|M_y - M_{y'}\|}, \mathcal{M}_y).$$

**Remark 4.9** (**How permutation affects generalization**). Although simplex ETF is the structure in the ideal neural collapse phenomenon, in practical implementations, it is hard for a model to achieve absolute collapse with an exact simplex ETF, which means $\gamma_{y,y'}$ cannot be equal for all $y \neq y'$ (we demonstrate empirically that margins of different class pairs have a large variance in the classification tasks in Appendix A.2), and $\|M_y - M_{y'}\|$ behaves similarly. Therefore, after permutation $\pi$ is performed on feature, $\hat{\gamma}_{y,y'}$ is not necessarily equal to $\hat{\gamma}_{\pi(y),\pi(y')}$, where we denote $\hat{\gamma}_{y,y'} = \frac{\gamma_{y,y'}}{L\|M_y - M_{y'}\|}$. Finally, this leads to the changed covering number $\mathcal{N}(\frac{\gamma_{\pi(y),\pi(y')}}{L\|M_y - M_{y'}\|}, \mathcal{M}_y)$, and affects the generalization bound. We provide an illustrative example in Figure 2 to explain how $\hat{\gamma}_{y,y'}$ affects the covering number.

The proof of Theorem 4.8 is based on interpolation and covering number complexity of the data space, and can provide the insight of why permutation can affect generalization performance. However, the dataset size it requires increases at an exponential rate with data dimension. Therefore, it is only applicable in low-dimensional data. We then further provide another theorem with an improved proof scheme, which not only relaxes the original assumptions, but also provides an error bound with faster convergence.

**Theorem 4.10.** *Given the balanced dataset $S$ and a classifier $(\boldsymbol{M}, f(\cdot; \boldsymbol{w}))$, suppose $(\boldsymbol{M}, f(\cdot; \boldsymbol{w}))$ can linearly separate $S$ by margin $\{\gamma_{y,y'}\}_{y \neq y'}$. Assume the maximum norm of features in $y$-th class is $\rho_y = \sup_{\boldsymbol{w}, \boldsymbol{x} \in \mathbb{P}_{x|y}} \|f(\boldsymbol{x}; \boldsymbol{w})\|$. Then the expected accuracy of $(\boldsymbol{M}, f(\cdot; \boldsymbol{w}))$ is given by*

$$Acc \geq 1 - \frac{2d}{C} \sum_{y=1}^{C} \left( \mathcal{H}\left(1, d, \rho_y, N/C\right) + \mathcal{H}\left(\min_{y' \neq y} \frac{\gamma_{y,y'}}{\|M_y - M_{y'}\|} - \sqrt{N/C}, d, \rho_y, N/C\right) \right)$$

*where we denote $\mathcal{H}(\alpha, d, \rho, n) = \exp\left(\frac{-n\alpha^2}{8d^2\rho^2}\right)$.*

**Remark 4.11.** The proof of Theorem 4.10 follows the same route to Theorem 4.8. The main difference is that Theorem 4.10 uses Hoeffding's Inequality (Hoeffding, 1994) to perform probability concentration on the feature learning of the model, instead of the interpolation-based technique. Through a similar mechanism, this theorem can also indicate that permutation can impact generalization, but is based on fewer and weaker assumptions, because it relaxes the assumptions of Lipschitz property and the large size of dataset and only assumes there is a maximum norm $\rho_y$ of feature in each class. Meanwhile, the error in Theorem 4.10 converges faster, which decays with the sample size $N$ at an exponential rate, while the original one in Theorem 4.8 is $\mathcal{O}(\frac{1}{N})$. Interestingly, it also reveals that the magnitude of features, $\rho_y$, can also affect generalization. An excessive magnitude can impair the classifier's generalization capacity.

Except permutation transformation of simplex ETF, previous work (Hao et al., 2015) points out that the rotation over the feature can impact the sparsity, which as a result, also potentially affects the generalization performance (Maurer et al., 2012; Hastie et al., 2015). Therefore, we conclude that the solution variability caused by permutation or rotation from different optimization paths leads to different generalization performance. We refer to this novel property as *"non-conservative generalization"*. In experiments, we find that models converging to equivalent simplex ETFs of different permutations and rotations show a large variance with respect to test loss and test accuracy, which empirically support the non-conservative generalization suggested by our theoretical results.

## 4.3 DISCUSSIONS

When it comes to the reason behind non-conservative generalization, it may be the optimization path. Training with different random factors, such as the initialization, may go through different optimization paths and converge in different label alignment and feature direction, although all of them can present the NC properties. This gives us the insight that there are some favored optimization paths, which can be beneficial to representation learning since their directed label alignment (permutation) and feature direction (rotation) can exploit an inductive bias (Goyal & Bengio, 2022), and consequently help to learn high-quality representation more easily. Meanwhile, there are also bad ones that may conflict with the inductive bias. For example, label alignment can be related to a widely used inductive bias in contrastive learning (Tian et al., 2020; Chen et al., 2020; Wang & Isola, 2020): the images that have closer semantic knowledge should be encoded closer in the representation space. Thus it is reasonable for us to align cat closer to dog in the representation space, rather

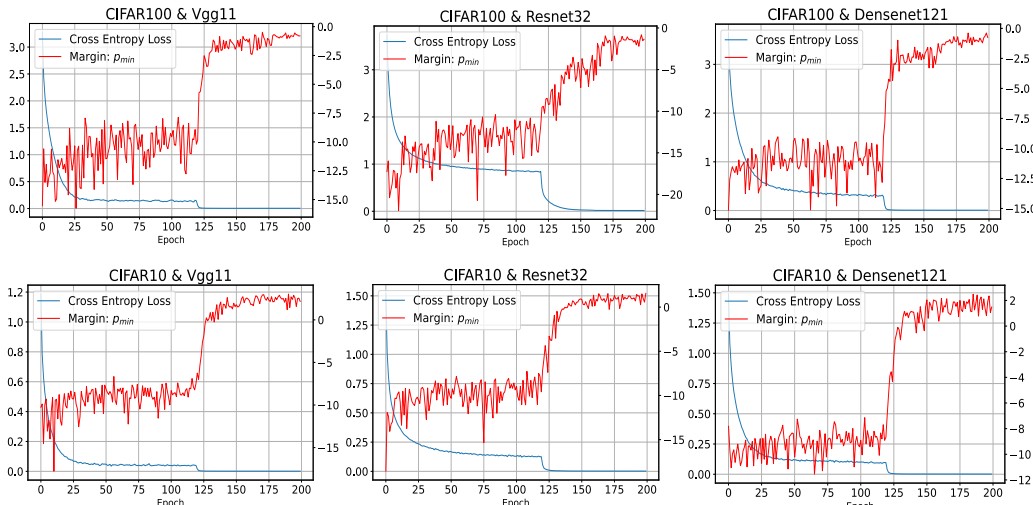

Figure 3: The CE loss and $p_{min}$ on the train set of VGG11, ResNet32 and DenseNet121 on CI-FAR10/100, respectively. The Y-axis on the left side specifies CE loss, and the right side specifies the margin $p_{min}$.

than the vehicle, to achieve a smaller margin between cat and dog than the margin between cat and vehicle, because cat has much more similar appearances with dog, but is completely different from vehicle. Therefore, through the lens of neural collapse, how to specify a rotation and permutation such that the model training can be performed along the right optimization path for better exploiting inductive bias deserves future exploration.

## 5 EXPERIMENTS

### 5.1 VERIFICATINON OF THE INCREASING MARGIN DURING TPT

The experiments in this part are established to validate the Theorem 1. As shown in Figure 3, the margin $p_{min}$ keeps increasing as the the CE loss is approximating zero. The setting and super parameters in the experiments of Figure 3 are the same as Section 5.2.

### 5.2 VERIFICATION OF THE NON-CONSERVATIVE GENERALIZATION

To investigate the impact of rotation and permutation transformations of simplex ETF on the generalization performance of deep neural networks, we conducted a series of experiments. The comprehensive results are illustrated in Figure 4.

**Implementation details.** The details of our experiments can be found in Appendix A.1.

**How to reveal the non-conservative generalization.** For a classification problem, we first generate a simplex ETF $M^\star \in \mathbb{R}^{d \times C}$ before training. and randomly generate 10 permutation matrices $\{P_i \in \mathbb{R}^{C \times C}\}_{i=1}^{10}$ and rotation matrices $\{R_i \in \mathbb{R}^{d \times d}\}_{i=1}^{10}$. Then, we train the model 10 times using the equivalent simplex ETF. To ensure the model would learn a expected simplex ETF, we follow the approach in Yang et al. (2022). They point out in their Theorem.1: if the linear classifier is fixed as simplex ETF, then the final features learned by the model would converge to be Simplex ETF with the same direction to classifier. In each time, we initialize the linear classifier as the equivalent simplex ETF $R_i M^\star$ or $M^\star P_i$, and do not perform optimization on it during training. To exclude the impact of the randomness factors, such as mini-batch, augmentation and parameter initialization, we use the same random seed for each training of 10 times. Once the NC phenomenon occurs, we know that the 10 models have learned Equivalent simplex ETFs. Finally, we compare their generalization performances by evaluating the CE loss and accuracy on a test set.

**The category number v.s. feature dimension.** To cover a more general case, we also provide experimental results when class number $C$ is larger than feature dimension $d$. Since the optimal

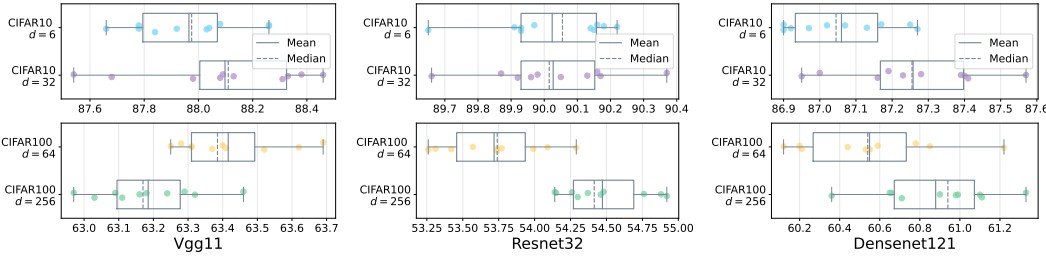

(a) Test accuracy comparison with different permutation transform.

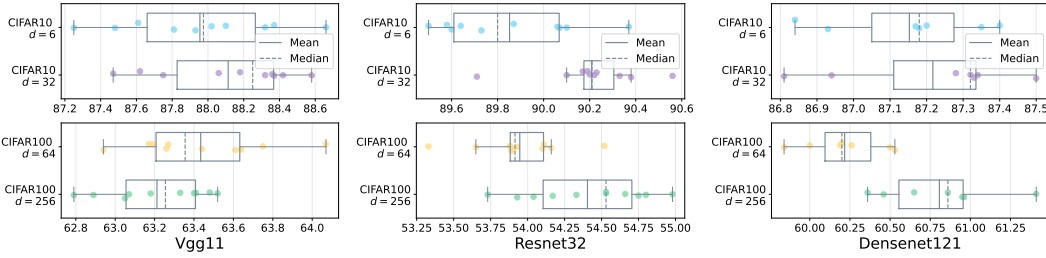

(b) Test accuracy comparison with different rotation transform.

Figure 4: Sensitivity analysis of test accuracy metric for different rotation and permutation on CIFAR10/100 and three different backbones, ResNet32, VGG11 and DenseNet121.

structure of NC in that case remains unclear [1], we obtain the structure by optimizing CE loss on the unconstrained feature models. Other experimental details when $C > d$ are completely the same to the case when $d >= C$.

**Results.** Figure 4 presents the comparisons of test accuracy metric, where the metrics in each box comes from 10 times training with the same random seed and super parameters. The only difference between them is that they have different initial permutation and rotation. All metrics are recorded when the classification model converges to NC ($100\%$ accuracy and zero loss on training set), which can ensure that the representation learning of models have reach their corresponding equivalent simplex ETF. We observe that, even though they achieve perfect performance on the train set, they still exhibit significant differences in test CE loss (shown in Figure 5 in Appendix A.3) and accuracy. These experimental results demonstrates that different feature alignment to label (permutation) and feature direction (rotation) can influence generalization capacity of models.

## 6 CONCLUSION

In this paper, we explore the generalization behaviors of classification models during the occurrence of neural collapse, We find the minimization of CE loss can be approximately seen as a multi-class SVM, which leads to increasing margins between every two classes during the terminal phase of training (TPT) when neural collapse emerges. Then based on our proposed multi-class margin generalization bound, we point out that a larger margin enjoys a stronger generalization capacity, which theoretically explains the test performance improvement observed during TPT. In addition, we find that the solution variability with different permutations can change the terms in the generalization error bound and consequently lead to different generalization performance. Finally, we validate our theoretical findings in experiments, showing that the permutation and rotation transforms both cause a large variance of test performance for models with 100% train accuracy, which verifies the non-conservation generalization property revealed by our theoretical results.

---

[1]Note that simplex ETF only exists when the number of class $C$ is smaller than feature dimension $d$ (refer to Definition 3.1). Most of existing studies about the optimal structure in NC only justify simplex ETF when $C \leq d$, without considering the case that $C > d$.

## STATEMENT

### ETHICS STATEMENT

We can ensure all authors adhere to the ICLR Code of Ethics. And our study does NOT involve any concerns about potential conflicts, biased practices, and other problems about public health, privacy, fairness, security, dishonest behaviors, *etc*.

### REPRODUCIBILITY STATEMENT

Our study is reproducible and does not involve novel models or algorithms. Clear explanations for theoretical results in this paper, including all assumptions and proof of claims, are documented in the appendix. Finally, a comprehensive description of data processing steps and other experimental details for all datasets is available in the appendix.

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
