

(a) Test cross-entropy loss with different permutation transform.

(b) Test cross-entropy loss with different rotation transform.

Figure 5: Sensitivity analysis of test CE metric for different rotation and permutation on CIFAR10/100 and three different backbones, ResNet32, VGG11 and Denset121.

# A    EXPERIMENTS

## A.1    MORE DETAILS ABOUT EXPERIMENTS

**Network Architecture and Dataset** Our experiments involve two image classification datasets: CIFAR10/100 Krizhevsky (2009). And for every dataset, we use three different convolutional neural networks to verify our finding, including ResNet He et al. (2016), VGG Simonyan & Zisserman (2015), DenseNet Huang et al. (2017). Both datasets are balanced with 10 and 100 classes respectively, each having 500 and 5,000 training images per class. We attach a linear layer after the end of backbone, which can transform feature dimensions. For CIFAR10, we use 32 and 6 as the feature dimensions of backbone. And for CIFAR100, we use 256 and 64 as the feature dimensions of backbone.

**Training** To reach NC phenomenon during training, we follow Papyan et al. (2020)'s practice. For all experiments, we minimize cross entropy loss using stochastic gradient descent with epoch 200, momentum 0.9, batch size 256 and weight decay $5 \times 10^{-4}$. Besides, the learning rate is set as $5 \times 10^{-2}$ and annealed by ten-fold at 120-th and 160-th epoch for every dataset. As for data preprocess, we only perform standard pixel-wise mean subtracting and deviation dividing on images. To achieve 100% accuracy on training set, we remove all dropout layers in the backbone and only perform RandomFilp augmentation.

## A.2    LARGE VARIANCE OF PAIR-WISE MARGIN

In the experiments of Section 5, we also records margins between each class pair. We record them at the epoch that model has the maximal test accuracy during TPT. The Figure 6 illustrates the all margins in the training of three backbones on the CIFAR10. We could find that these margins exhibit large variability, which means that the feature of classification model does not exhibit the rigorous Simplex ETF structure. We can observe that the difference between margins are still much large even during TPT, which provide empirical support for our conclusion in Remark 4.9.

### A.3 MORE METRIC COMPARISON

We also provide test CE loss comparison to verify the non-conservative generalization, which is illustrated in Figure 5.

## B PROOF OF THEOREM 4.1

**Theorem 4.1** (**Multiclass SVM**). *For the CE loss function (1), consider a path of gradient flow* $\{(\boldsymbol{M}^{(t)}, \boldsymbol{Z}^{(t)})\}_t$. *If* $\text{CELoss}(\boldsymbol{M}^{(t)}, \boldsymbol{Z}^{(t)}) \to 0$ ($t \to \infty$), *then* $p_{min} \to \infty$ ($t \to \infty$), *where* $p_{min}$ *is the margin of the train set*

$$p_{min} := \min_{y \neq y'} \min_{i \in [N/C]} \langle M_y - M_{y'}, z_{y,i} \rangle.$$

*Proof.* For simplicity, we leave out the upper script $(t)$. First, we have $\forall t$

$$
\begin{aligned}
\text{CELoss}(\boldsymbol{Z}, \boldsymbol{M}) &= \sum_{y=1}^{C} \sum_{i=1}^{N/C} -\log \frac{\exp\left(\langle M_y, \boldsymbol{z}_{y,i} \rangle\right)}{\sum_{y'} \exp\left(\langle M_{y'}, \boldsymbol{z}_{y,i} \rangle\right)} \\
&= \sum_{y=1}^{C} \sum_{i=1}^{N/C} \log\left(1 + \sum_{y' \neq y} \exp\left(\langle M_{y'} - M_y, \boldsymbol{z}_{y,i} \rangle\right)\right) \\
&\leq \sum_{y=1}^{C} \sum_{i=1}^{N/C} \log\left(1 + (C-1) \exp\left(\max_{y' \neq y}\{\langle M_{y'} - M_y, \boldsymbol{z}_{y,i} \rangle\}\right)\right) \\
&\leq \frac{N}{C} \sum_{y=1}^{C} \log\left(1 + (C-1) \exp\left(\max_{y' \neq y} \max_{i \in [N/C]}\{\langle M_{y'} - M_y, \boldsymbol{z}_{y,i} \rangle\}\right)\right) \\
&\leq N \max_{y \in [C]} \log\left(1 + (C-1) \exp\left(\max_{y' \neq y} \max_{i \in [N/C]}\{\langle M_{y'} - M_y, \boldsymbol{z}_{y,i} \rangle\}\right)\right) \\
&= N \log\left(1 + (C-1) \exp\left(\max_{y \in [C]} \max_{y' \neq y} \max_{i \in [N/C]}\{\langle M_{y'} - M_y, \boldsymbol{z}_{y,i} \rangle\}\right)\right)
\end{aligned}
\tag{2}
$$

In addition, we have

$$\log\left(1 + \exp\left(\max_{y \in [C]} \max_{y' \neq y} \max_{i \in [N/C]}\{\langle M_{y'} - M_y, \boldsymbol{z}_{y,i} \rangle\}\right)\right) \leq \text{CELoss}(\boldsymbol{Z}, \boldsymbol{M}) \tag{3}$$

We denote $\max_{y \in [C]} \max_{y' \neq y}$ as $\max_{y' \neq y}$ and define the margin of entire dataset (refer to Section.3.1 of Ji et al. (2022)) as follow:

$$p_{min} := \min_{y \neq y'} \min_{i \in [N/C]} \langle M_y - M_{y'}, z_{y,i} \rangle$$

Therefore, we have

$$\underbrace{\log\left(1 + \exp\left(-p_{min}\right)\right)}_{\ell_1(p_{min})} \leq \text{CELoss}(\boldsymbol{Z}, \boldsymbol{M}) \leq N \underbrace{\log\left(1 + (C-1) \exp\left(-p_{min}\right)\right)}_{\ell_{C-1}(p_{min})} \tag{4}$$

where $\ell_a(p) = \log(1 + ae^{-p})$. Then we represent $\ell_a(\cdot)$ as the form of exponential function, *i.e.*

$$\ell_a(p) = e^{-\phi_a(p)} \quad \text{and} \quad \phi_a(p) = -\log\log(1 + ae^{-p}).$$

Denote the inverse function of $\phi_a(\cdot)$ as $\Phi_a(\cdot)$, where $\Phi_a(p) = -\log(\frac{e^{e^{-p}} - 1}{a})$. Then continue from (4), we have

$$
\begin{aligned}
&\ell_1(p_{min}) \leq \text{CELoss}(\boldsymbol{Z}, \boldsymbol{M}) \leq N\ell_{C-1}(p_{min}) \\
\Leftrightarrow &e^{-\phi_1(p_{min})} \leq \text{CELoss}(\boldsymbol{Z}, \boldsymbol{M}) \leq Ne^{-\phi_{C-1}(p_{min})} \\
\Leftrightarrow &\phi_{C-1}(p_{min}) - \log(N) \leq -\log(\text{CELoss}(\boldsymbol{Z}, \boldsymbol{M})) \leq \phi_1(p_{min})
\end{aligned}
$$

According to the monotonicity of $\Phi_1(\cdot)$, we have

$$\Phi_1\left(\phi_{C-1}(p_{min}) - \log(N)\right) \leq \Phi_1\left(-\log(\text{CELoss}(\boldsymbol{Z}, \boldsymbol{M}))\right) \leq p_{min}$$

Use the mean value theorem, there exists a $\xi \in (\phi_{C-1}(p_{min}) - \log(N), \phi_1(p_{min}))$ such that

$$\Phi_1(\phi_{C-1}(p_{min}) - \log(N)) = p_{min} - \Phi_1^{'}(\xi)(\phi_1(p_{min}) - \phi_{C-1}(p_{min}) + \log(N)),$$

then

$$p_{min} - \underbrace{\Phi_1^{'}(\xi)(\phi_1(p_{min}) - \phi_{C-1}(p_{min}) + \log(N))}_{\Delta(t)} \leq \Phi_1\left(-\log(\text{CELoss}(\boldsymbol{Z}, \boldsymbol{M}))\right) \leq p_{min} \quad (5)$$

Then we will show $\Delta(t) = \mathcal{O}(1)(t \to \infty)$. Since

$$\xi > \phi_{C-1}(p_{min}) - \log(N) \geq -\log(\text{CELoss}(\boldsymbol{Z}, \boldsymbol{M})) - \log(N),$$

we know $\xi \to \infty$ and $p_{min} \to \infty$ as $\text{CELoss}(\boldsymbol{Z}, \boldsymbol{M}) \to 0$. By simple calculation, we have $\phi_1(p_{min}) - \phi_{C-1}(p_{min}) \to \log(C-1)$ and $\Phi_1^{'}(\xi) = \frac{e^{e^{-\xi}-\xi}}{e^{e^{-\xi}}-1} \to 1$. Therefore, as $\text{CELoss}(\boldsymbol{Z}, \boldsymbol{M}) \to 0$, we have $p_{min} \to +\infty$ according to (5) and monotonicity of $\Phi_1(-\log(\cdot))$. $\qquad\square$

## C   PROOF OF THEOREM 4.4

The following lemma provide a two-class classification generalization bound based on margin.

**Lemma C.1 (Theorem.5 of Kakade et al. (2008): Margin Bound).** *Consider a data space $\mathcal{X}$ and a probability measure $\mathcal{P}$ on it. There is a dataset $\{x_i\}_{i=1}^n$ that contains $n$ samples, which are drawn i.i.d from $\mathcal{P}$. Consider an arbitrary function class $\mathcal{F}$ such that $\forall f \in \mathcal{F}$ we have $\sup_{x \in \mathcal{X}}|f(x)| \leq K$, then with probability at least $1 - \delta$ over the sample, for all margins $\gamma > 0$ and all $f \in \mathcal{F}$ we have,*

$$\mathbb{P}_x(f(x) \leq 0) \leq \sum_{i=1}^n \frac{\mathbb{I}(f(x_i) \leq \gamma)}{n} + \frac{\mathfrak{R}_n(\mathcal{F})}{\gamma} + \sqrt{\frac{\log(\log_2\frac{4K}{\gamma})}{n}} + \sqrt{\frac{\log(1/\delta)}{2n}}$$

We give a multiclass version of Lemma C.1.

**Theorem 4.4 (Multiclass Margin Bound).** *Consider a dataset $S$ with $C$ classes. For any classifier $(\boldsymbol{M}, f(\cdot; \boldsymbol{w}))$, we denote its margin between $y$ and $y'$ classes as $(M_y - M_{y'})^T f(\cdot; \boldsymbol{w})$. And suppose the function space of the margin is $\mathcal{F} = \{(M_y - M_{y'})^T f(\cdot; \boldsymbol{w})|\forall y \neq y', \forall \boldsymbol{M}, \boldsymbol{w}\}$, whose uppder bound is*

$$\sup_{y \neq y'} \sup_{\boldsymbol{M}, \boldsymbol{w}} \sup_{x \in \mathcal{M}_y} \left|(M_y - M_{y'})^T f(\boldsymbol{x}; \boldsymbol{w})\right| \leq K.$$

*Then, for any classifier $(\boldsymbol{M}, f(\cdot; \boldsymbol{w}))$ and margins $\{\gamma_{y,y'}\}_{y \neq y'} (\gamma_{y,y'} > 0)$, the following inequality holds with probability at least $1 - \delta$*

$$\mathbb{P}_{x,y}\left(\max_{y'}[Mf(\boldsymbol{x}; \boldsymbol{w})]_{y'} \neq y\right) \leq \sum_{y=1}^C p(y) \sum_{y' \neq y} \frac{\mathfrak{R}_{N_y}(\mathcal{F})}{\gamma_{y,y'}} + \sum_{y=1}^C p(y) \sum_{y' \neq y} \sqrt{\frac{\log(\log_2\frac{4K}{\gamma_{y,y'}})}{N_y}}$$
$$+ \; \text{empirical risk term} \; + \; \text{probability term}$$

*where*

$$\text{empirical risk term} = \sum_{y=1}^C p(y) \sum_{y' \neq y} \sum_{x \in S_y} \frac{\mathbb{I}((M_y - M_{y'})^T f(x) \leq \gamma_{y,y'})}{N_y},$$
$$\text{probability term} = \sum_{y=1}^C p(y) \sum_{y' \neq y} \sqrt{\frac{\log(C(C-1)/\delta)}{2N_y}}.$$

*$\mathfrak{R}_{N_y}(\mathcal{F})$ is the Rademacher complexity Kakade et al. (2008); Bartlett & Mendelson (2002) of function space $\mathcal{F}$.*

*Proof.* We decompose the error as errors within every class by Bayes Theory:

$$\mathbb{P}_{\boldsymbol{x},y}\Big(\arg\max_{y'}[\boldsymbol{M}f(\boldsymbol{x};\boldsymbol{w})]_{y'}\neq y\Big) = \sum_{y=1}^{C} p(y)\mathbb{P}_{\boldsymbol{x}|y}\Big(\arg\max_{y'}[\boldsymbol{M}f(\boldsymbol{x};\boldsymbol{w})]_{y'}\neq y\Big) \qquad (6)$$

where $p(y)$ is the probability density of $y$-th class. Then, we focus on the accuracy within every class $y$.

$$\mathbb{P}_{\boldsymbol{x}|y}\Big(\arg\max_{y'}[\boldsymbol{M}f(\boldsymbol{x};\boldsymbol{w})]_{y'}\neq y\Big) = \mathbb{P}_{\boldsymbol{x}|y}\Big(\bigcup_{y'\neq y}\{(M_y - M_{y'})^T f(\boldsymbol{x};\boldsymbol{w}) < 0\}\Big)$$

According to union bound, we have

$$\mathbb{P}_{\boldsymbol{x}|y}\Big(\arg\max_{y'}[\boldsymbol{M}f(\boldsymbol{x};\boldsymbol{w})]_{y'}\neq y\Big) \leq \sum_{y'\neq y}\mathbb{P}_{\boldsymbol{x}|y}\Big((M_y - M_{y'})^T f(\boldsymbol{x};\boldsymbol{w}) < 0\Big)$$

Recall our assumption of function class:

$$\sup_{y\neq y'}\sup_{\boldsymbol{M},\boldsymbol{w}}\sup_{\boldsymbol{x}\in\mathcal{M}_y}|(M_y - M_{y'})^T f(\boldsymbol{x};\boldsymbol{w})| \leq K.$$

Then follow from the Margin Bound (Theorem C.1), we have

$$\mathbb{P}_{\boldsymbol{x},y}\Big(\arg\max_{y'}[Mf(\boldsymbol{x};\boldsymbol{w})]_{y'}\neq y\Big) \leq \sum_{y=1}^{C}p(y)\sum_{y'\neq y}\mathbb{P}_{x|y}\Big((M_y - M_{y'})^T f(\boldsymbol{x};\boldsymbol{w}) < 0\Big)$$

$$\leq \sum_{y=1}^{C}p(y)\sum_{y'\neq y}\frac{\Re_{N_y}(\mathcal{F})}{\gamma_{y,y'}} + \sum_{y=1}^{C}p(y)\sum_{y'\neq y}\sqrt{\frac{\log(\log_2\frac{4K}{\gamma_{y,y'}})}{N_y}} +$$

$$\underbrace{\sum_{y=1}^{C}p(y)\sum_{y'\neq y}\sqrt{\frac{\log(1/\delta)}{2N_y}}}_{\text{probability term}} + \underbrace{\sum_{y=1}^{C}p(y)\sum_{y'\neq y}\sum_{\boldsymbol{x}\in S_y}\frac{\mathbb{I}((M_y - M_{y'})^T f(\boldsymbol{x}) \leq \gamma_{y,y'})}{N_y}}_{\text{empirical risk term}}$$

with probability at least $1 - C(C-1)\delta$. Then, we perform the following replace to drive the final result:

$$\delta \leftarrow \frac{\delta}{C(C-1)}$$

$\square$

## D  PROOF OF THEOREM 4.8

We first introduce the definition of Covering Number.

**Definition D.1** (**Covering Number** Kulkarni & Posner (1995))**.** Given $\epsilon > 0$ and $\boldsymbol{x}\in\mathbb{R}^D$, the open ball of radius $\epsilon$ around $\boldsymbol{x}$ is denoted as

$$B_\epsilon(\boldsymbol{x}) = \{\boldsymbol{u}\in\mathbb{R}^D, \|\boldsymbol{u}-\boldsymbol{x}\| < \epsilon\}.$$

Then the covering number $\mathcal{N}(\epsilon, A)$ of a set $A \subset \mathbb{R}^D$ is defined as the smallest number of open balls whose union contains $A$:

$$\mathcal{N}(\epsilon, A) = \inf\left\{k : \exists \boldsymbol{u}_1,\ldots,\boldsymbol{u}_k\in\mathbb{R}^D, s.t. A \in \bigcup_{i=1}^{k}B_\epsilon(\boldsymbol{u}_i)\right\}$$

The following conclusion is demonstrated in the proof of Theorem.1 of Kulkarni & Posner (1995). We use it to prove our theorem.

**Lemma D.2** (Vural & Guillemot (2017); Kulkarni & Posner (1995)). *There are $N$ samples $\{x_1, \ldots, x_N\}$ drawn i.i.d from the probability measure $\mathcal{P}$. Suppose the bounded support of $\mathcal{P}$ is $\mathcal{M}$, then if $N$ is larger then Covering Number $\mathcal{N}(\epsilon, \mathcal{M})$, we have*

$$\mathbb{P}_x\Big(\|x - \hat{x}\| > \epsilon\Big) \leq \frac{\mathcal{N}(\epsilon, \mathcal{M})}{2N}, \forall \epsilon > 0$$

*where $\hat{x}$ is the sample that is closest to $x$ in $\{x_1, \ldots, x_N\}$:*

$$\hat{x} \in \underset{x' \in \{x_1, \ldots, x_N\}}{\arg\min} \|x' - x\|$$

Then we provide the proof of Theorem 4.8.

**Theorem 4.8.** *Given a balanced dataset $S$ and a classifier $(\boldsymbol{M}, f(\cdot; \boldsymbol{w}))$, suppose $(\boldsymbol{M}, f(\cdot; \boldsymbol{w}))$ can linearly separate $S$ by margin $\{\gamma_{y,y'}\}_{y \neq y'}$. Besides, we make the following assumptions:*

- *$f(\cdot, \boldsymbol{w})$ is $L$-Lipschitz for any $\boldsymbol{w}$, i.e. $\forall \boldsymbol{x}_1, \boldsymbol{x}_2, \|f(\boldsymbol{x}_1, \boldsymbol{w}) - f(\boldsymbol{x}_2, \boldsymbol{w})\| \leq L\|\boldsymbol{x}_1 - \boldsymbol{x}_2\|$*

- *$S$ is large enough such that $N_y \geq \max_{y' \neq y} \mathcal{N}(\frac{\gamma_{y,y'}}{L\|M_y - M_{y'}\|}, \mathcal{M}_y)$ for every class $y$*

- *The tight support of probability $\mathcal{P}_{\boldsymbol{x}|y}$ is denoted as $\mathcal{M}_y$*

*where $\mathcal{N}(\cdot, \mathcal{M}_y)$ is the covering number of $\mathcal{M}_y$. Please refer to Appendix D for its definition. Then the expected accuracy of $(\boldsymbol{M}, f(\cdot; \boldsymbol{w}))$ over the entire distribution is given by*

$$\mathbb{P}_{\boldsymbol{x},y}\Big(\max_{y'}[Mf(\boldsymbol{x}; \boldsymbol{w})]_{y'} = y\Big) > 1 - \frac{1}{2N}\sum_{y=1}^{C}\max_{y' \neq y}\mathcal{N}(\frac{\gamma_{y,y'}}{L\|M_y - M_{y'}\|}, \mathcal{M}_y).$$

*Proof.* We decompose the accuracy:

$$\mathbb{P}_{\boldsymbol{x},y}\Big(\max_{y'}[Mf(\boldsymbol{x}; \boldsymbol{w})]_{y'} = y\Big) = \sum_{y=1}^{C} p(y)\mathbb{P}_{\boldsymbol{x}|y}\Big(\max_{y'}[Mf(\boldsymbol{x}; \boldsymbol{w})]_{y'} = y\Big) \tag{7}$$

where $p(y)$ is the class distribution. Then, we focus on the error within every class $i$.

$$\mathbb{P}_{\boldsymbol{x}|y}(\max_{y'}[Mf(\boldsymbol{x}; \boldsymbol{w})]_{y'} = y) = \mathbb{P}_{x|y}(\{(M_y - M_{y'})^T f(\boldsymbol{x}; \boldsymbol{w}) > 0 \text{ for any } y' \neq y\})$$

We select the data that is closest to $\boldsymbol{x}$ in $y$ class samples $S_y$, and denote it as

$$\hat{\boldsymbol{x}}(S_i) = \underset{\boldsymbol{x}_1 \in S_i}{\arg\min}\|\boldsymbol{x}_1 - x\|$$

According to the linear separability,

$$(M_y - M_{y'})^T f(\hat{\boldsymbol{x}}(S_y); \boldsymbol{w}) \geq \gamma_{y,y'}, \forall y' \neq y$$

For any $y' \neq y$, we have

$$\begin{aligned}
(M_y - M_{y'})^T f(\boldsymbol{x}; \boldsymbol{w}) &= (M_y - M_{y'})^T (f(\boldsymbol{x}; \boldsymbol{w}) + f(\hat{\boldsymbol{x}}(S_y); w) - f(\hat{\boldsymbol{x}}(S_y); w)) \\
&= (M_y - M_{y'})^T f(\hat{\boldsymbol{x}}(S_y); w) + (M_y - M_{y'})^T (f(\boldsymbol{x}; \boldsymbol{w}) - f(\hat{\boldsymbol{x}}(S_y); w)) \\
&\geq \gamma_{y,y'} - \|M_y - M_{y'}\|\|f(\boldsymbol{x}; \boldsymbol{w}) - f(\hat{\boldsymbol{x}}(S_y); w)\| \\
&\geq \gamma_{y,y'} - L\|M_y - M_{y'}\|\|\boldsymbol{x} - \hat{\boldsymbol{x}}(S_y)\|
\end{aligned} \tag{8}$$

The prediction result is related to the distance between $\boldsymbol{x}$ and $\hat{\boldsymbol{x}}(S_y)$. According to Theorem D.2, we know

$$\mathbb{P}_{\boldsymbol{x}|y}\Big(\|\boldsymbol{x} - \hat{\boldsymbol{x}}(S_y)\| > \epsilon\Big) \leq \frac{\mathcal{N}(\epsilon, \mathcal{M}_y)}{2N_y}$$

To obtain the correct prediction result, *i.e.*, assure (8) $> 0$ for all $y' \neq y$, we choose $\epsilon < \min_{y' \neq y} \frac{\gamma_{y,y'}}{L\|M_y - M_{y'}\|}$. Therefore, we have

$$
\mathbb{P}_{\boldsymbol{x}|y}\left( \left\{(M_y - M_{y'})^T f(\boldsymbol{x};\boldsymbol{w}) > 0, \forall y' \neq y\right\} \right) \geq \mathbb{P}_{\boldsymbol{x}|y}\left( \|\boldsymbol{x} - \hat{\boldsymbol{x}}(S_y)\| < \min_{y' \neq y} \frac{\gamma_{y,y'}}{L\|M_y - M_{y'}\|} \right)
$$
$$
> 1 - \frac{\mathcal{N}(\min_{y' \neq y} \frac{\gamma_{y,y'}}{L\|M_y - M_{y'}\|}, \mathcal{M}_y)}{2N_y}
$$

(9)

Plug (9) into (7) to derive

$$
\mathbb{P}_{\boldsymbol{x},y}\left( \max_{y'}[Mf(\boldsymbol{x};\boldsymbol{w})]_{y'} = y \right) > 1 - \sum_{y=1}^{C} p(y)\frac{\mathcal{N}(\min_{y' \neq y} \frac{\gamma_{y,y'}}{L\|M_y - M_{y'}\|}, \mathcal{M}_y)}{2N_y}
$$
$$
= 1 - \sum_{y=1}^{C} p(y)\frac{\max_{y' \neq y} \mathcal{N}(\frac{\gamma_{y,y'}}{L\|M_y - M_{y'}\|}, \mathcal{M}_y)}{2N_y}
$$

$\square$

# E  PROOF OF THEOREM 4.10

First, we provide the Hoeffding's Inequality Mohri et al. (2012).

**Lemma E.1** (Hoeffding's Inequality). *Let $X_1, \ldots, X_n$ be independent random variables such that $b \leq X_i \leq a$. Consider the sum of these random variables, let $\hat{E}_n(X) = \frac{1}{n}\sum_{i=1}^{n} X_i$, then $\forall \epsilon > 0$, we have*

$$
\mathbb{P}\left( \hat{E}_n(X) - \mathbb{E}(\hat{E}_n(X)) \geq \epsilon \right) \leq \exp\left(-2n\epsilon^2/(b-a)^2\right)
$$
$$
\mathbb{P}\left( \mathbb{E}(\hat{E}_n(X)) - \hat{E}_n(X) \geq \epsilon \right) \leq \exp\left(-2n\epsilon^2/(b-a)^2\right)
$$
$$
\mathbb{P}\left( \left|\hat{E}_n(X) - \mathbb{E}(\hat{E}_n(X))\right| \geq \epsilon \right) \leq 2\exp\left(-2n\epsilon^2/(b-a)^2\right)
$$

Then, we extend the Hoeffding's Inequality to a higher-dimensional form.

**Lemma E.2** (High Dimensional Hoeffding's Inequality). *Let $\boldsymbol{X}_1, \ldots, \boldsymbol{X}_n$ be independent random vectors $\in \mathbb{R}^D$ such that $\|\boldsymbol{X}_i\| \leq \rho, \forall i$, where $\boldsymbol{X}_i^j$ indecates the $j$-th coordinate of $\boldsymbol{X}_i$. Consider the sum of them, let $\hat{E}_n(\boldsymbol{X}) = \frac{1}{n}\sum_{i=1}^{n} \boldsymbol{X}_i$, then $\forall \epsilon > 0$, we have*

$$
\mathbb{P}\left( \left\|\hat{E}_n(\boldsymbol{X}) - \mathbb{E}\left(\hat{E}(\boldsymbol{X})\right)\right\| \geq \epsilon \right) \leq 2D\exp\left(-n\epsilon^2/2D^2\rho^2\right)
$$

*Proof.* Due to $\|\boldsymbol{X}_i\| \leq \rho \ (\forall i)$, we know $-\rho \leq \boldsymbol{X}_i^j \leq \rho \ (\forall i, j)$. According to Lemma E.1, we know $\forall j \in [D]$,

$$
\mathbb{P}\left( \left|\hat{E}_n(\boldsymbol{X}^j) - \mathbb{E}\left(\hat{E}(\boldsymbol{X}^j)\right)\right| \geq \epsilon \right) \leq 2\exp\left(-n\epsilon^2/2\rho^2\right),
$$
$$
\text{where } \hat{E}_n(\boldsymbol{X}^j) = \frac{1}{n}\sum_{i=1}^{n} \boldsymbol{X}_i^j
$$

then

$$
\mathbb{P}\left( \left|\hat{E}_n(\boldsymbol{X}^j) - \mathbb{E}\left(\hat{E}(\boldsymbol{X}^j)\right)\right| < \epsilon \right) > 1 - 2\exp\left(-n\epsilon^2/2\rho^2\right),
$$

and we combine all $j \in [D]$,

$$
\mathbb{P}\left( \sum_{j=1}^{D} \left|\hat{E}_n(\boldsymbol{X}^j) - \mathbb{E}\left(\hat{E}(\boldsymbol{X}^j)\right)\right| < D\epsilon \right) > 1 - 2D\exp\left(-n\epsilon^2/2\rho^2\right)
$$

Then, we perform the $\epsilon \leftarrow \frac{\epsilon}{D}$. And according to the following formula, we drive the final result.

$$\left\| \hat{E}_n(\boldsymbol{X}) - \mathbb{E}\left(\hat{E}(\boldsymbol{X})\right) \right\| \leq \sum_{j=1}^{D} \left| \hat{E}_n(\boldsymbol{X}^j) - \mathbb{E}\left(\hat{E}(\boldsymbol{X}^j)\right) \right|,$$

$\square$

**Corollary E.3.** *Let $\boldsymbol{X}_1, \ldots, \boldsymbol{X}_n$ be i.i.d random vectors $\in \mathbb{R}^D$ such that $\mathbb{E}(\boldsymbol{X}_i) = \mathbb{E}(\boldsymbol{X}), \forall i$ and $-\rho \leq \boldsymbol{X}_i^j \leq \rho, \forall j$, where $\boldsymbol{X}_i^j$ indecates the $j$-th coordinate of $\boldsymbol{X}_i$. Consider the sum of them, let $\hat{E}_n(\boldsymbol{X}) = \frac{1}{n}\sum_{i=1}^n \boldsymbol{X}_i$, then $\forall \epsilon > 0$, we have*

$$\mathbb{P}\left(\left\|\hat{E}_n(\boldsymbol{X}) - \mathbb{E}(\boldsymbol{X})\right\| \geq \epsilon\right) \leq 2D\exp\left(-n\epsilon^2/2D^2\rho^2\right)$$

In our settings, all samples in the $y$-th class are i.i.d from probability $\mathcal{P}_{\boldsymbol{x}|y}$. So we apply the Corollary.E.3 in the proof of Theorem 4.10.

**Theorem 4.10.** *Given the balanced dataset $S$ and a classifier $(\boldsymbol{M}, f(\cdot; \boldsymbol{w}))$, suppose $(\boldsymbol{M}, f(\cdot; \boldsymbol{w}))$ can linearly separate $S$ by margin $\{\gamma_{y,y'}\}_{y\neq y'}$. Assume the maximum norm of features in $y$-th class is $\rho_y = \sup_{\boldsymbol{w}, \boldsymbol{x} \in \mathbb{P}_{\boldsymbol{x}|y}} \|f(\boldsymbol{x}; \boldsymbol{w})\|$. Then the expected accuracy of $(\boldsymbol{M}, f(\cdot; \boldsymbol{w}))$ is given by*

$$Acc \geq 1 - \frac{2d}{C}\sum_{y=1}^{C}\left(\mathcal{H}\left(1, d, \rho_y, N/C\right) + \mathcal{H}\left(\min_{y'\neq y}\frac{\gamma_{y,y'}}{\|M_y - M_{y'}\|} - \sqrt{N/C}, d, \rho_y, N/C\right)\right)$$

*where we denote $\mathcal{H}(\alpha, d, \rho, n) = \exp\left(\frac{-n\alpha^2}{8d^2\rho^2}\right)$.*

*Proof.* We denote the class center of $y$-th class as $\hat{\boldsymbol{\mu}}_y = \frac{1}{N_y}\sum_{\boldsymbol{x} \in S_y} f(\boldsymbol{x}; \boldsymbol{w})$ and $\boldsymbol{\mu} = \mathbb{E}_{\boldsymbol{x}|y}\boldsymbol{x} = \mathbb{E}_{S_y}\hat{\boldsymbol{\mu}}_y$. and start from (8) in the proof of Theorem 4.10.

$$\begin{aligned}
(M_y - M_{y'})^T f(\boldsymbol{x}; \boldsymbol{w}) &= (M_y - M_{y'})^T (f(\boldsymbol{x}; \boldsymbol{w}) + \hat{\boldsymbol{\mu}}_y - \hat{\boldsymbol{\mu}}_y) \\
&= (M_y - M_{y'})^T \hat{\boldsymbol{\mu}}_y + (M_y - M_{y'})^T (f(\boldsymbol{x}; \boldsymbol{w}) - \hat{\boldsymbol{\mu}}_y) \\
&\geq \gamma_{y,y'} - \|M_y - M_{y'}\|\|f(\boldsymbol{x}; \boldsymbol{w}) - \hat{\boldsymbol{\mu}}_y\| \\
&\geq \gamma_{y,y'} - \|M_y - M_{y'}\| (\|f(\boldsymbol{x}; \boldsymbol{w}) - \boldsymbol{\mu}_y\| + \|\boldsymbol{\mu}_y - \hat{\boldsymbol{\mu}}_y\|)
\end{aligned} \quad (10)$$

According to Corollary.E.3, we have $\forall \delta \in (0, 1)$

$$\mathbb{P}_{\boldsymbol{x}|y}\left(\|f(\boldsymbol{x}; \boldsymbol{w}) - \boldsymbol{\mu}_y\| \geq \epsilon\right) \leq 2d\exp\left(-\epsilon^2/2d^2\rho_y^2\right)$$

$$\mathbb{P}_{S_y}\left(\|\hat{\boldsymbol{\mu}}_y - \boldsymbol{\mu}_y\| \geq \epsilon\right) \leq 2d\exp\left(-n\epsilon^2/2d^2\rho_y^2\right)$$

Let $\epsilon = \frac{1}{2}\min_{y'\neq y}\frac{\gamma_{y,y'}}{\|M_y - M_{y'}\|}$, then

$$\mathbb{P}_{\boldsymbol{x}|y, S_y}\left(\left\{(M_y - M_{y'})^T f(\boldsymbol{x}; \boldsymbol{w}) > 0, \forall y' \neq y\right\}\right)$$

$$\geq \mathbb{P}_{\boldsymbol{x}|y, S_y}\left(\|f(\boldsymbol{x}; \boldsymbol{w}) - \boldsymbol{\mu}_y\| + \|\hat{\boldsymbol{\mu}}_y - \boldsymbol{\mu}_y\| \geq \min_{y'\neq y}\frac{\gamma_{y,y'}}{\|M_y - M_{y'}\|}\right)$$

$$\geq 1 - \mathbb{P}_{\boldsymbol{x}|y, S_y}\left(\|f(\boldsymbol{x}; \boldsymbol{w}) - \boldsymbol{\mu}_y\| \geq \sqrt{N_y}\right) - \mathbb{P}_{\boldsymbol{x}|y, S_y}\left(\|\hat{\boldsymbol{\mu}}_y - \boldsymbol{\mu}_y\| \geq \min_{y'\neq y}\frac{\gamma_{y,y'}}{\|M_y - M_{y'}\|} - \sqrt{N_y}\right)$$

$$\geq 1 - 2d\exp\left(\frac{-N_y}{8d^2\rho_y^2}\right) - 2d\exp\left(\frac{-N_y\left(\min_{y'\neq y}\frac{\gamma_{y,y'}}{\|M_y - M_{y'}\|} - \sqrt{N_y}\right)^2}{8d^2\rho_y^2}\right)$$

So

$$\mathbb{P}_{\boldsymbol{x}, y, S}\left(\arg\max_{y'}\left[Mf(\boldsymbol{x}; \boldsymbol{w})\right]_{y'} = y\right)$$

$$\geq 1 - \frac{2d}{C}\sum_{y=1}^{C}\left(\mathcal{H}\left(1, d, \rho_y, N_y\right) + \mathcal{H}\left(\min_{y'\neq y}\frac{\gamma_{y,y'}}{\|M_y - M_{y'}\|} - \sqrt{N_y}, d, \rho_y, N_y\right)\right)$$

$\square$

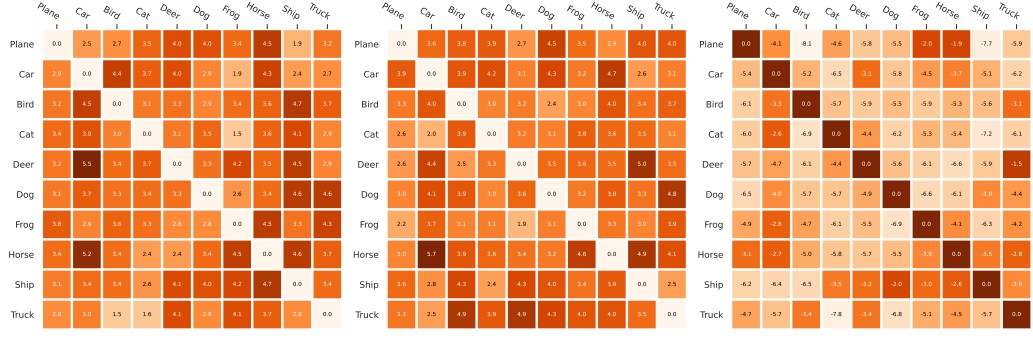

(a) Pair-wise margins comparison with three different permutations on DenseNet121.

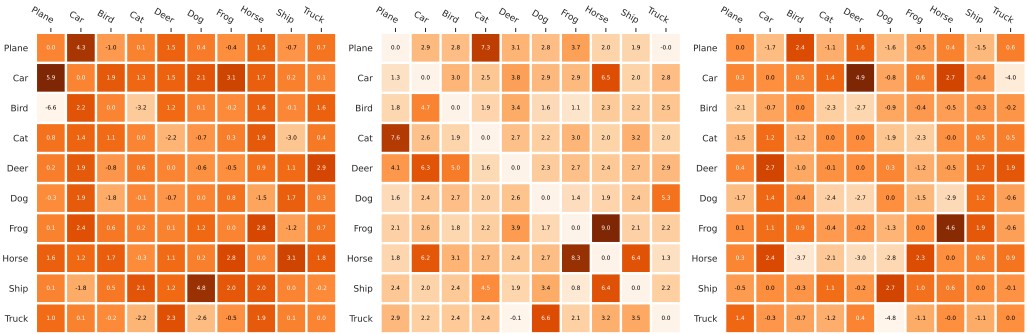

(b) Pair-wise margins comparison with three different permutations on ResNet32.

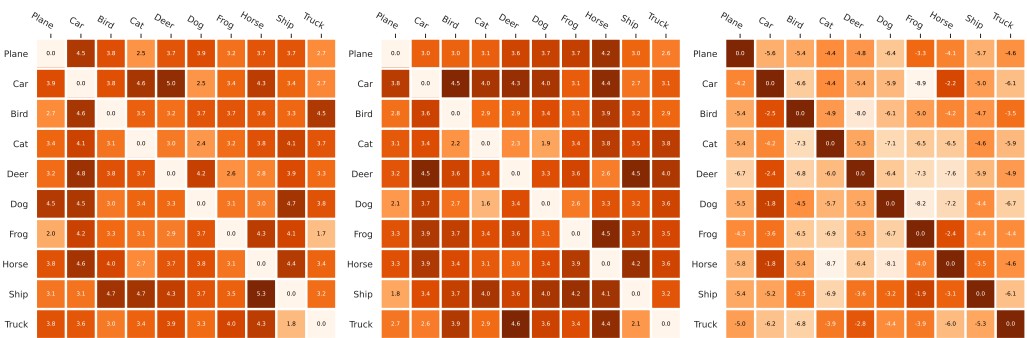

(c) Pair-wise margins comparison with three different rotations on DenseNet121.

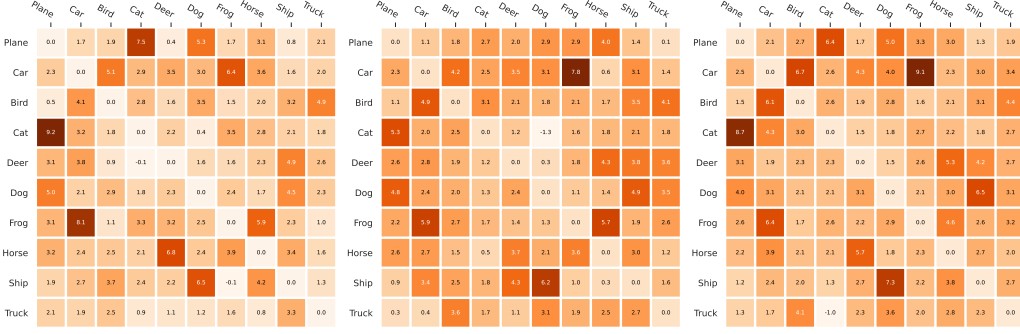

(d) Pair-wise margins comparison with three different rotations on ResNet32.

Figure 6: Values comparison of pair-wise margins on training of ResNet32 and DenseNet121 with different permutations and rotations on CIFAR10.