# OpenReview forum: "Towards Demystifying the Generalization Behaviors When Neural Collapse Emerges"
_ICLR.cc/2024/Conference — ICLR 2024 Conference Withdrawn Submission_

### Official Review · Reviewer_PCNB · 2023-10-29

**Soundness:** 2 fair
**Presentation:** 3 good
**Contribution:** 1 poor
**Rating:** 3
**Confidence:** 5

**Summary:**

This paper delves into the generalization behavior during NC, specifically the intriguing phenomenon of continued improvement in generalization accuracy even after the training accuracy reaches 100%. The authors establish a connection between cross-entropy minimization and a multi-class SVM during TPT, deriving a multi-class margin generalization bound to theoretically explain this continued improvement. Furthermore, the research uncovers the concept of "non-conservative generalization," highlighting that different alignments between labels and features in a simplex ETF can lead to varying degrees of generalization improvement, despite all models reaching NC and displaying similar training performance. Empirical experiments confirm the findings and insights provided by the theoretical results.

**Strengths:**

- The paper is well-written and easy to follow.

**Weaknesses:**

- The results in this paper, derived while analyzing the cross-entropy (CE) loss and involving SVM, exhibit similarities to those in Ji et al. (2021) [1]. The authors should provide a more detailed comparative analysis. Additionally, there is a significant flaw in Theorem 4.1, where it is possible to expand $p_{min}$ by merely scaling $M$ or $z$. The authors do not analyze the relationship between the norms of $M$ (or $z$) and $p_min$, and the relevant experiments are not reflected. This deviation from SVM's behavior is not explained with reference to relevant literature.
- The multi-class generalization bound proposed by the authors is not novel, as similar conclusions have been proposed or discussed in the works of Cao et al. [2] and Zhou et al. [3] Furthermore, these prior works also explore the concept that enlarging the margin $\gamma$ can lead to improved generalization. Consequently, the contributions in this regard appear to be less significant and should be more thoroughly supported with references to related literature.
- The author suggests that permutations can affect generalization; however, the variations in test accuracy depicted in Figure 4 do not exhibit pronounced differences (the term "large variance" is not an accurate characterization). On the other hand, the author links the impact of permutations on generalization to $\|M_y-M_{y'}\|$, suggesting that fluctuations in the norms of linear classifiers for each category may be causing changes in generalization error. The author needs to conduct more extensive experiments on this point, especially considering the possibility of eliminating these effects through normalization.
- The experimental results in Figure 4 may not accurately reflect real-world outcomes, as the associated test accuracy is lower than typical performance benchmarks. For instance, the highest test accuracy on CIFAR-100 is only about 64%. The authors should conduct more extensive experiements on a variety of dataset and under more common settings to provide a broader perspective.

[1] Ji W, Lu Y, Zhang Y, et al. An unconstrained layer-peeled perspective on neural collapse[J]. arXiv preprint arXiv:2110.02796, 2021.

[2] Cao K, Wei C, Gaidon A, et al. Learning imbalanced datasets with label-distribution-aware margin loss[J]. Advances in neural information processing systems, 2019, 32.

[3] Zhou, X., Liu, X., Zhai, D., Jiang, J., Gao, X., & Ji, X. Learning towards the largest margins. *arXiv preprint arXiv:2206.11589*.

**Questions:**

as mentioned in the weakness part

---

### Official Review · Reviewer_LrEp · 2023-10-30

**Soundness:** 3 good
**Presentation:** 3 good
**Contribution:** 2 fair
**Rating:** 5
**Confidence:** 4

**Summary:**

In this paper, the authors establish the connection between the minimization of CE and a multi-class SVM during TPT, and then they derive a multi-class margin generalization bound. This provides a theoretical explanation for why continuing training can still lead to accuracy improvement on test set. Additionally, further theoretical results are provided (the “non-conservative generalization” property) which indicate that different alignment between labels and features in a simplex ETF can result in varying degrees of generalization improvement. Finally, the authors provide empirical observations to verify the indications suggested by the theoretical results.

**Strengths:**

This paper analyze the generalization behaviors when neural collapse emerges, and I find two points that are interesting.

1.  by looking at the minimization of the CE (cross-entropy) loss in the last layer during TPT (terminal phase of training) to a multi-class SVM, the authors show that during TPT the margin keeps increasing while the CE loss gets smaller.

2. the proof scheme inspired by out-of-sample interpolation requires less assumption and the empirical results show that permutations and rotations of different solutions leads to different generalization performance.

Those two parts provide new "metrics" for studying the correlation of neural collapse and generalization.

**Weaknesses:**

1. The idea that "The minimization of CE loss can be seen as a multi-class SVM" is not very new, see [1]. Then the novelty of multi-class margin generalization bound and the connection between margin and generalization performance becomes less.

2. As mentioned in the paper, what are good permutations and rotations for good generalization, which is one important question, is not answered in the paper.


[1] Wang, Ke, Vidya Muthukumar, and Christos Thrampoulidis. "Benign overfitting in multiclass classification: All roads lead to interpolation." Advances in Neural Information Processing Systems 34 (2021): 24164-24179.

**Questions:**

The paper is clearly written and I have no questions.

---

### Official Review · Reviewer_c2cC · 2023-10-30

**Soundness:** 3 good
**Presentation:** 3 good
**Contribution:** 2 fair
**Rating:** 5
**Confidence:** 3

**Summary:**

The study delves into the generalization capabilities of models trained for classification tasks, providing insights into the phenomenon where a model's generalization performances continue to improve after entering the Terminal Phase Training (TPT) where Neural Collapse occurs. The paper utilizes an unconstrained feature model and casts the original problem to a multiclass SVM problem for analysis and further demonstrates that permutation/rotation of the Simplex ETF structure can lead to different generalization performance. Experiments are conducted across various network architectures and datasets to support the claim in the paper.

**Strengths:**

1. The paper is well organized and the topics covered are very interesting.
2. The transformation of the original optimization problem to a multiclass SVM problem is natural and insightful.
3. The investigation of how the permutation and rotation of the Simplex ETF influence test accuracy is interesting and can potentially motivate future research.

**Weaknesses:**

1. At its foundation, the approach of examining generalization within the context of the unconstrained feature model appears counterintuitive to the reviewer. This is primarily because the unconstrained feature model omits the inputs, making it challenging to discuss generalization without incorporating input in the analysis. Furthermore, the majority of the theoretical results listed in the paper focus on the training regime, directly linking such results to generalization is not totally convincing to the reviewer.
2. In Theorem 4.8, the authors assume the feature extractor to be L-Lipschitz, this is highly unlikely in most realistic networks (consider the fact that modern networks are notoriously weak under adversarial attack), which further weakens the applicability of this theorem.

**Questions:**

1. Theorem 4.10 depends on less strict assumptions and has a better convergence rate than Theorem 4.8, is there any practical benefit of Theorem 4.8 on top of Theorem 4.10? Because otherwise why not just delete Theorem 4.8?
2. In Figure 3, it seems like $p_{min}$ is smaller than 0 for the CIFAR-100 dataset even at the end of the training, does this mean the training accuracy is not reaching 100%?
3. In Section 5.2, the paper talks about the influence of permutation/rotation of the Simplex ETF on the test results, is there an optimal configuration, perhaps a specific permutation or rotation, that consistently delivers the highest test accuracy?

---

### Official Review · Reviewer_zHE3 · 2023-10-31

**Soundness:** 2 fair
**Presentation:** 3 good
**Contribution:** 1 poor
**Rating:** 1
**Confidence:** 4

**Summary:**

This paper analyzes generalization in deep networks in the setting of neural collapse, attempting to explain why generalization improves while training beyond zero training error, and why different models with approximate neural collapse can exhibit different generalization properties. For the first question, the authors use elements of margin theory and Rademacher complexity analysis to bound the generalization error as a function of the margin obtained in the last layer. For the latter, they use recent results on supervised manifold  learning based on covering numbers. The authors complement their results with some empirical verification.

**Strengths:**

* This paper studies a fundamental problem that has not yet been addressed by the community: on the one hand, there are plenty of (not very successful) generalization results for deep neural networks, but none of them are able to explain the neural collapse phenomenon; on the other hand, we know that neural collapse is a ubiquitous phenomenon in modern networks. From this perspective, this work could be very important and timely.

* The paper is very clearly written, provides very good background, and it was fun to read.

**Weaknesses:**

First, I should stress that I read this paper with great enthusiasm and excitement for the results that it promised in the title and abstract. However, it is my opinion that the biggest weaknesses of this paper are the soundness of their main results. I expand on this at length below but, in nutshell, some are simple corollaries of known results, others are new but unable to convey any useful information, and others have potentially some important technical problems. I'm hoping that my understanding is amiss, and that the authors can correct my observations below. For the same reason, don't put much (or any) importance on my rating of the paper for now, as this will completely depend on the authors' responses and discussions, which I look forward to.

**Questions:**

1. **Theorem 4.1 and implications**

Th. 4.1 shows that, for the unconstrained feature model (Zhu et al, 2021), trained with cross entropy loss (Eq 1), if the loss tends to 0, then the margin on the training set tends to infinity.

This result is very similar to previous results (Soudry et al, 2018; Ji & Telgarsky 2019) which show that gradient descent on separable data converges to a max margin classifier and to the global minimum of the training problem. Note: these results provide a much tighter analysis as they provide convergence rates, also. The results in 4.1 is not exactly the same, as here the authors talk about gradient flow instead of gradient descent, but I would happily argue that the latter is significantly more relevant as it is what is actually computed in practice. Furthermore, note that the only reason that the margin in Thm 4.1 diverges to infinity is because the authors consider the un-normalized margin, and the classifier diverge in norm. Thus, in light of these previous results, I don't see what novelty Th4.1 provides.

2. **Theorem 4.4 and implications**

This result, which is one of the two main results of this paper, provides a bound on the risk of the classifier $F(x) = \arg\max M f(x,w)$, where $f(x,w)$ is a representation computed by some deep network. The bound depends on the empirical margin risk, as well as a generalization gap bound based on the Radamacher complexity of the function class (plus a term that ensure that the bound holds uniformly for any margin). In particular, this gap is $\tilde{\mathcal O}(\mathfrak R(\mathcal F))/\gamma$, where $\gamma$ is the required/observed margin.

I have a few comments on this result:
* i) The hypothesis/function class is defined as $\mathcal F$ is defined for all $M$ and $w$ in their respected spaces. Thus, as defined, $K$ is unbounded (unless $f$ is trivial/useless) because this allows for $||M||\to\infty$, or for $||f(x,w)||\to\infty$ because either $||w||\to\infty$ or even because $||x||\to\infty$ (since we don't even know if $\mathcal X$ is unbounded). So, all of these should be put in place.

* ii) A main concern I have is that this results is nothing else than a standard margin bound; see e.g. Theorem 5.9. in (Mohi et al, "Foundations of Machine Learning", 2018). The only difference here is the muti-class setting, but these are also standard/direct extensions.

* iii) A second main concern with this results is that, while the characterization of the margin in the last layer is fine (and, as per the previous point, known), everything naturally hinges on the Radamacher complexity of the function class, $\mathfrak R(\mathcal F)$. Now,  in their Remark 4.6, the authors explain that this result (Thm 4.4, or Corollary 4.5: ``The reason for generalization improvement when NC emerges") explains the improvements in generalization because the margin increases (i.e. $\gamma_{i,j}\geq p_{min}$), thus decreasing the term $\tilde{\mathcal O}(\mathfrak R(\mathcal F))/\gamma$. However, in this setting of bounded parameter spaces, the margin cannot increase indefinitely (as it can in the unconstrained feature model/linear regression setting of Thm 4.1). Moreover, the increase in margin will naturally come at the expense of increasing the complexity of the function class, i.e. increasing $\tilde{\mathcal O}(\mathfrak R(\mathcal F))$. This trade-off is not knew, and indeed all results that use margin theory to provide generalization bounds for deep nets rely on characterizing and controlling this trade-off most effectively. Such a trade-off depends typically on the different norms of the parameters $w$; the authors might want to look at refs (A,B) in detail. As a result, without telling us how $\tilde{\mathcal O}(\mathfrak R(\mathcal F))$ varies with the obtained $\gamma$, this results says nothing new and it certainly does not explain or demystify why networks generalize well during neural collapse.

3. **Theorem 4.8**

This theorem attempts to provide a generalization bound relying on the covering number of the support of the data for a Lipschitz continuous $f(\cdot,w)$, basically providing $\mathbb P (\text{missclassification}) \leq \frac{1}{2N}\sum_y \max_{y'\not y}\mathcal N(\epsilon,\mathcal M_y)$ where the resolution $\epsilon = \gamma_{y,y'}/(L||M_y - M_{y'}||)$.

* i) Again, $L$ might be unbounded if the hypothesis space is unbounded.

* ii) In remark 4.9 the authors attempt to explain how this results might be able to explain why different models with similar collapse have the different risks. In particular, the authors write that permuting $\hat\gamma_{i,j}$ to $\hat\gamma_{\pi(i),\pi(j)}$, which might differ under approximate collapse, might result in different upper bounds on the risk. This is interesting, but the proposed covering number (as presented) is not computable and, as the authors mention, the number of data points required would depend exponentially on the dimension.

4. **Theorem 4.10**

In order to alleviate the shortcomings of Thm 4.8, here the authors provide an improved results which seems to show that the risk decays exponentially fast with the number of samples.

* i) Again, note that $\rho$ might be unbounded.

* ii) My main observation here is that I don't think this result is correct: the authors employ a tail bound (Hoeffding's) on the learned features but, unfortunately, Hoeffding's requires samples to be i.i.d for concentration. More specifically, looking at their proof, page 19, they write:

$\mathbb P_{S_y}( || \hat{\mu}_y - \mu_y || \geq \epsilon ) \leq 2d \exp(-n\epsilon^2 / 2d^2\rho^2_y)$

where

$\hat{\mu_y}  = \frac{1}{N_y} \sum_{x_j \in S_y} f(x_j,w)$ and $\mu_y = \mathbb E \hat{\mu}_y$.

However, the samples $f(x_j,w)$ that are empirically averaged need to be independent, and they *are not* independent because $w$ depends on the sample $S$. More generally, this is the reason why concentration bounds are not sufficient to provide generalization bounds.

---

References:

A: Bartlett, Peter L., Dylan J. Foster, and Matus J. Telgarsky. "Spectrally-normalized margin bounds for neural networks." Advances in neural information processing systems 30 (2017).

B: Wei, Colin, and Tengyu Ma. "Improved sample complexities for deep neural networks and robust classification via an all-layer margin." International Conference on Learning Representations. 2019.

---

**Minor comments**
* CE in abstract is undefined.
* first sentence of second paragraph: "physics" -> "mechanisms"? there's really no physics here.
* Thm 4.4: "uppder" -> "upper".
* Thm 4.8: what does "tight" support mean? Did you mean bounded?
* Right before the discussions, the authors mention an interesting connection to sparsity and its role in generalization. I wonder if the authors believe that their intuition could be related to the work in (Muthukumar et al. "Sparsity-aware generalization theory for deep neural networks." COLT, 2023)